# Deep Learning-Based Identification of Collapsed, Non-Collapsed and Blue Tarp-Covered Buildings from Post-Disaster Aerial Images

**Hiroyuki Miura [1],* , Tomohiro Aridome [2] and Masashi Matsuoka [3]**

[1] Department of Advanced Science and Engineering, Hiroshima University, Kagamiyama 1-4-1, Higashi-Hiroshima, Hiroshima 739-8527, Japan
[2] Department of Architecture, Hiroshima University, Kagamiyama 1-4-1, Higashi-Hiroshima, Hiroshima 739-8527, Japan; tomo.jawm@gmail.com
[3] Department of Architecture and Building Engineering, Tokyo Institute of Technology, Nagatsuta 4259, Yokohama, Kanagawa 226-8502, Japan; matsuoka.m.ab@m.titech.ac.jp
* Correspondence: hmiura@hiroshima-u.ac.jp; Tel.: +81-82-424-7798

**Abstract:** A methodology for the automated identification of building damage from post-disaster aerial images was developed based on convolutional neural network (CNN) and building damage inventories. The aerial images and the building damage data obtained in the 2016 Kumamoto, and the 1995 Kobe, Japan earthquakes were analyzed. Since the roofs of many moderately damaged houses are covered with blue tarps immediately after disasters, not only collapsed and non-collapsed buildings but also the buildings covered with blue tarps were identified by the proposed method. The CNN architecture developed in this study correctly classifies the building damage with the accuracy of approximately 95 % in both earthquake data. We applied the developed CNN model to aerial images in Chiba, Japan, damaged by the typhoon in September 2019. The result shows that more than 90 % of the building damage are correctly classified by the CNN model.

**Keywords:** deep learning; building damage; aerial image; earthquake; typhoon

---

## 1. Introduction

Large-scale natural disasters, such as earthquakes, have produced a huge number of building damage over a wide area. In order to consider the effective emergency response and early-stage recovery planning, it is indispensable to identify the amount and distribution of the structural damage immediately after the disaster. Remote sensing has been recognized as a suitable source to provide timely data for the detection of building damage in large areas (e.g., [1,2]). Although visual interpretations have been applied to detect building damage from optical remote-sensing images [3–5], a lot of time and human resources are required to identify the damage over a wide area. Gray level or texture-based change detection techniques between pre- and post-event images have been introduced in order to automatically or semi-automatically estimate damage distributions [6–11]. Such change-based approaches, however, cannot be applied when a pre-event image is not available in the area of interest. A robust methodology to automatically detect building damage only from post-disaster images needs to be developed for rapid and effective damage assessment.

Recently, deep-learning and artificial intelligence solutions have been dramatically developing in the various fields of science and engineering. Convolutional neural network (CNN), one of the deep-learning techniques, is now recognized as the most suitable approach for image recognition (e.g., [12,13]). CNN-based approaches have been applied to remote sensing images not only for scene and land cover classifications [14,15] but also for detecting building damage areas in earthquake

disasters [16–23]. Duarte et al. [16] demonstrated the CNN-based feature fusion technique for identifying a damaged region from remote sensing images. Nex et al. [17] examined the transfer-learning technique for detecting the visible structural damage from satellites, airborne and UAV images. Ma et al. [18,19] accepted the CNN-based approaches for estimating the block-level damage distribution and extracting the damaged parts of buildings from satellite images, respectively. Cooner et al. [20] and Ji et al. [21,22] applied the machine- or deep-learning techniques for creating building-by-building damage maps by including building footprints in the image analysis. Since building footprint data are available for most urban areas from open databases such as the OpenStreetMap service [24], the use of building footprints becomes more feasible for easily delineating locations of individual buildings from an image.

These studies revealed that the CNN-based approaches have worked successfully in building damage detections by adjusting the CNN architectures suitably for their datasets. Most of the previous studies, however, classified the building damage into two classes; damaged and non-damaged or collapsed and non-collapsed. As pointed out in Ci et al. [23], the identification of an intermediate damage level would be important for actual post-disaster activities including early-stage recovery planning and disaster waste management. Moreover, most of the previous studies used damage data visually interpreted from remote-sensing images as reference training data. As pointed out in the previous validation studies of visual interpretations [3,25], uncertainties and/or mis-classifications were contained not only in intermediate damage levels but also in no damage and severe damage levels. In order to develop a reliable damage detection technique, ground truth data obtained in field damage investigations need to be used as reference training data.

When building damage is identified from vertically observed remote sensing images, a building roof is an important factor for classifying the damage grade because the damage of building sides such as walls and columns, and the internal damage, cannot be directly observed from the images. Since a building roof is one of the most fragile parts in residential houses, residents immediately cover the building roofs with blue plastic tarps to temporally protect their houses from further damage, such as water leakage, when the roofs are damaged by natural disasters. Although it is difficult to determine the detailed damage grade of the buildings covered by blue tarps from remote-sensing images, the blue tarps on building roofs themselves can be recognized as a sign of damage. Such blue tarp-covered buildings, however, were rarely considered explicitly in remote-sensing-based damage detection studies except in limited cases [26,27].

In this study, a CNN-based building damage detection technique was developed to automatically classify the damage grade into collapsed, non-collapsed and blue tarp-covered buildings by analyzing post-disaster aerial images and building damage inventories in the 2016 Kumamoto and the 1995 Kobe, Japan, earthquakes. The building damage inventories obtained in the field investigations by structural engineers were mainly used as reference ground truth data. Blue tarp-covered buildings were visually identified from the images, and the damage grades of the blue tarp buildings were discussed based on the damage inventories. The image patches were extracted from the post-disaster images with the help of the building footprints in the damage inventories. The CNN-based damage classification model was developed by the deep-learning of the damage grades of the buildings in the image patches. The applicability of the developed CNN architecture was discussed by estimating the building damage from the post-event aerial images observed in a town of the Chiba prefecture, Japan, damaged by the typhoon in September 2019.

## 2. Building Damage Inventories and Aerial Images

### 2.1. Damage Inventories of the 2016 Kumamoto, Japan Earthquake

In the Kumamoto prefecture of Kyushu Island, Japan (see Figure 1a), urban areas were severely damaged by the two large earthquakes that occurred on April 14 and 16, 2016, with moment magnitudes (Mw) of 6.2 and 7.0 (hereafter we refer to them as the 2016 Kumamoto earthquake). Figure 1b shows the distribution of the seismic intensities in the epicentral area during the Kumamoto earthquake

on 16 April, calculated from the observed strong motion data [28]. Since the seismic intensity, 7, the maximum seismic intensity level on the Japanese scale, was recorded during the earthquake in Matshiki town, Kumamoto prefecture, as shown in Figure 1b, a large number of buildings were severely damaged, especially in the town.

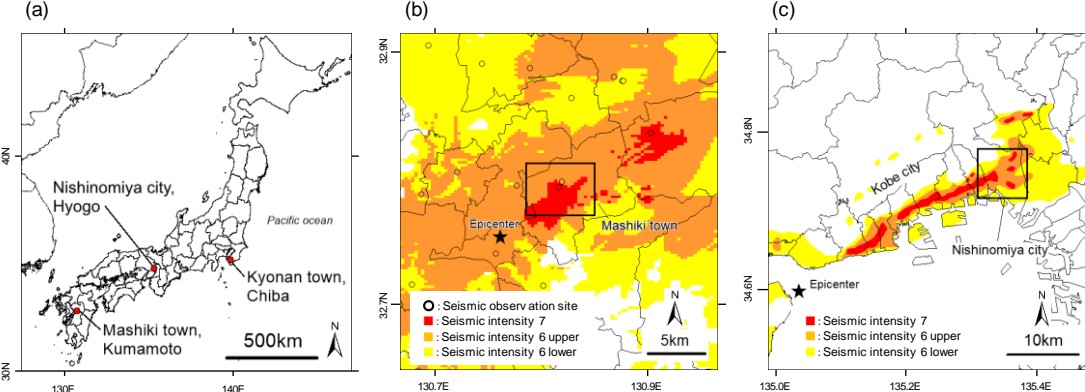

**Figure 1.** (**a**) Locations of the target areas of this study. (**b**) Seismic intensity map of the Kumamoto earthquake on 16 April 2016 [28], in and around Mashiki town, Kumamoto prefecture. (**c**) Seismic intensity map of the 1995 Kobe earthquake [29], in and around Nishinomiya city, Hyogo prefecture. Rectangle in (**b**) and (**c**) indicates the areas of interest.

After the earthquakes, the Kyushu branch of the Architectural Institute of Japan (AIJ) conducted the detailed building damage investigation in the central part of Mashiki town [30]. The damage grades proposed by Okada and Takai [31] were assigned to more than 2000 buildings by investigating the conditions of the buildings from the structural points of view in the field survey. Seven damage grades (here referred to as the AIJ-scale) from D0 to D6 were assigned to each building. Figure 2 illustrates the relationship of the damage grades between the European Macroseismic Scale (EMS-98) [32], globally used in building damage investigations, and the AIJ-scale. Typical descriptions for the building damage in EMS-98 and AIJ-scale are summarized in Table 1. Considering the illustrations and descriptions of the damage in Figure 2 and Table 1, the damage grades of D0 and D1 in the AIJ-scale almost correspond to negligible to slight damage (G1) in the EMS-98 because both D1 and G1 damage levels are characterized as hair-line cracks in walls. D5 and D6 in the AIJ-scale correspond to destruction damage (G5) in the EMS-98 because G5 includes not only complete collapse but near total collapse.

Figure 3a shows the distribution of the building damage in the central part of Mashiki town. The broken line area indicates the field survey area by the AIJ teams. Since the original damage inventory consists of point data, building footprints of the Geospatial Information Authority of Japan (GSI) [33] were given to the inventory. The building footprints included two structural classes; normal and substantial buildings. The normal buildings were defined as the timber buildings and the buildings lower than three-stories, and the substantial buildings were defined as the reinforced concrete and the steel structure buildings higher than three-stories [33]. Most of the normal buildings corresponded to one- or two-story residential houses in this area. The Japanese residential houses generally have gable roofs covered with tiles made from clay, cement, slate or metal materials. On the contrary, the substantial buildings have a flat roof slab constructed by concrete on the tops. Since the materials, structural types and failure mechanism of the substantial buildings are totally different from those of the residential houses, the substantial buildings were excluded from the analysis in the following steps. Hereafter, we refer to the damage inventory as the AIJ-data.

The building damage in the affected areas by the Kumamoto earthquake were also investigated by the visual interpretation of the post-event aerial images in Monma et al. [34]. They classified the damage grade of each building into four categories; major, heavy, moderate and negligible damage, based on the criteria in Table 1. Hereafter, we refer to these damage data as V (visual interpretation)-data. Considering the damage descriptions, the damage grades of the V-data can be approximated as shown

in Figure 2. The boundaries of the damage grades are not aligned with those in the AIJ-scale because of the difference of the criteria. The major damage almost corresponds to D6, D5 and some of the D4 buildings, and the negligible damage covers the D0 and some of the D1 buildings. The damage distribution outside of the field survey area in Figure 3a represents the result of the visual interpretation. In order to increase the number of the training data, the building damage inventory by the visual interpretation in Figure 3a is additionally used in this study. As mentioned above, the substantial buildings were also excluded in the following analysis.

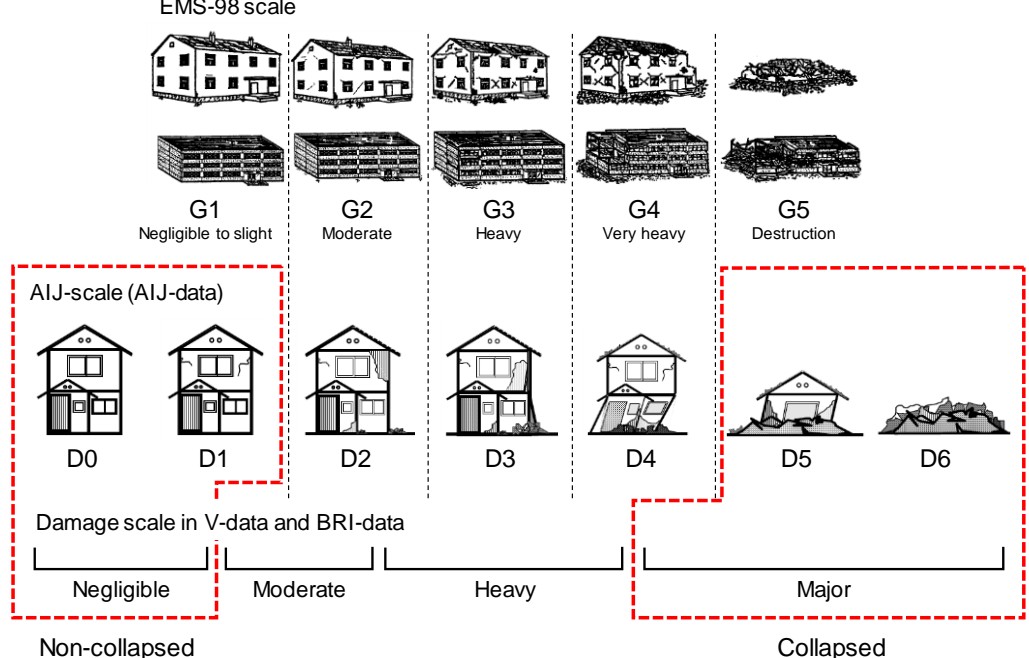

**Figure 2.** Comparison of the building damage scales of the Architectural Institute of Japan (AIJ)-scale [31], the European Macroseismic Scale (EMS-98) [32] and the damage scale in the V (visual interpretation)-data [34] and the Building Research Institute of Japan (BRI)-data [35]. D0, D1 and the negligible damage levels are classified into non-collapse, and D5, D6 and the major damage levels are classified into collapse in this study.

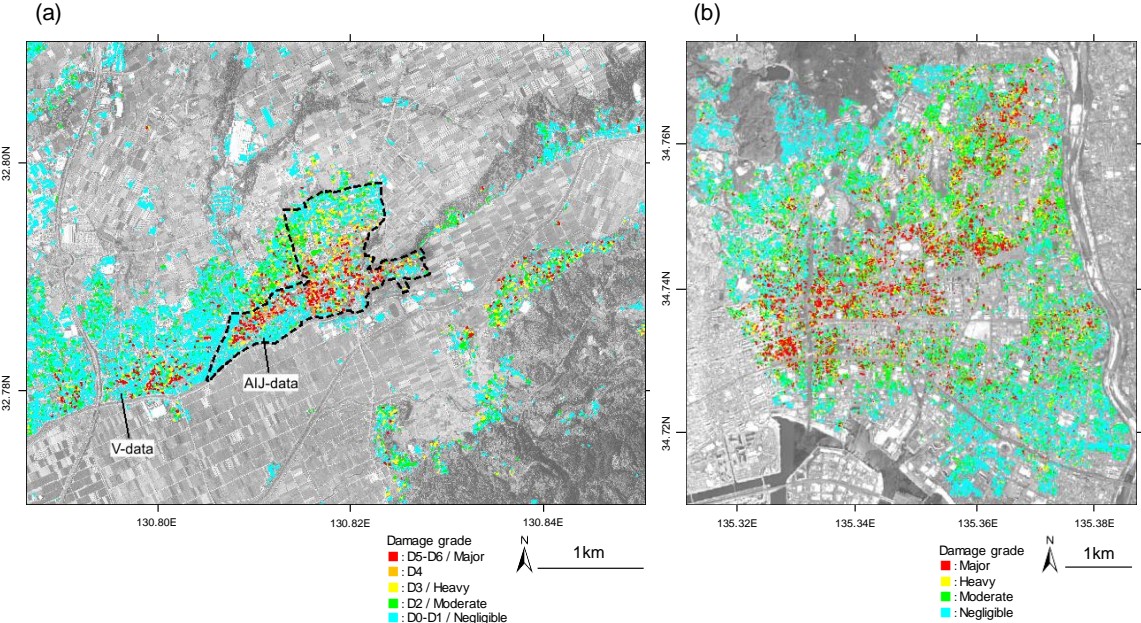

**Figure 3.** (**a**) Damage distribution of AIJ- and V-data in Mathiki town affected by the 2016 Kumamoto earthquake. (**b**) Damage distribution of BRI-data in Nishinomiya city affected by the 1995 Kobe earthquake.

**Table 1.** Descriptions of the building damage in EMS-98, AIJ-scale, V-data and BRI-data defined in the literature [31,32,34,35].

| Damage Scale/Damage Inventory | Damage Grade | Typical Damage Descriptions | Reference |
|---|---|---|---|
| EMS-98 | G1 | Hair-line cracks in very few walls; Fall of small pieces of plaster only | [32] |
| | G2 | Cracks in many walls; Falls of fairly large pieces of plaster | |
| | G3 | Large and extensive cracks in most walls; Roof tile detach; Failure of individual non-structural elements | |
| | G4 | Serious failure of walls; Partial structural failure of roofs and floors | |
| | G5 | Total or near total collapse | |
| AIJ-scale/AIJ-data | D0 | No damage | [31] |
| | D1 | Hair-line cracks in walls | |
| | D2 | Falls of plaster of walls; Detachment of roof tiles | |
| | D3 | Failure of columns; beams and walls; Serious failure of roof | |
| | D4 | Significant incline of columns; Serious failure of beams and walls | |
| | D5 | Partial collapse (Collapse of ground floor or upper floor) | |
| | D6 | Complete collapse | |
| V-data | Negligible | No visible damage | [34] |
| | Moderate | Detachment of roof tiles | |
| | Heavy | Serious failure of roof; Failure of walls | |
| | Major | Complete or partial collapse | |
| BRI-data | Negligible | No visible damage | [35] |
| | Moderate | Hair-line cracks in walls; Falls of plaster of walls; Detachment of roof tiles | |
| | Heavy | Incline of columns; Serious failure of roof; Significant cracks in walls | |
| | Major | Complete collapse; Significant incline of columns; Serious failure in columns and frames | |

## 2.2. Damage Inventory of the 1995 Kobe, Japan, Earthquake

The Kobe city area in Hyogo prefecture, Japan (see Figure 1a), was destructively damaged by a Mw 6.9 earthquake on 15 January 1995 (hereafter we refer to it as the 1995 Kobe earthquake). Nishinomiya city, Hyogo prefecture, was heavily damaged because the seismic intensity 7 was observed in the area as shown by the seismic intensity map [35] in Figure 1c. After the earthquake, the structural damage database was constructed in Nishinomiya city [36]. Among the building damage data included in the database, we analyzed the building damage inventory investigated by structural engineers of the AIJ and the City Planning Institute of Japan, and compiled by the Building Research Institute of Japan (BRI) [35]. They classified the damage grades into four levels (major, heavy, moderate and negligible damage) based on the criteria in Table 1. Hereafter, we refer the damage inventory as BRI-data. Although the criteria of the damage grades of the BRI-data are not perfectly consistent with those of the V-data, the major damage almost corresponds to the D5, D6 and some of the D4 buildings, and the negligible damage corresponds to the D0 and some of the D1 buildings in the AIJ-scale as shown in Figure 2. The damage inventory included building use such as residential, commercial and industrial in each polygon. Whereas the residential buildings almost corresponded to one- or two-story timber houses, most of the commercial and industrial buildings were constructed by reinforced concrete frames or steel frame structures. The residential houses were analyzed in the following steps by excluding the commercial and industrial buildings from the inventory because the materials, structural types and failure mechanism of those buildings were totally different from residential houses. Figure 3b shows the distribution of the damage for the residential houses in the 1995 Kobe earthquake.

The number of buildings in each damage inventory is summarized in Table 2. Approximately 2100, 8000 and 46,000 building data were included in the inventories, respectively. Approximately 15% of the buildings were classified as D5–D6/major damage in the AIJ-data and BRI-data. About 45–70% of the buildings were classified as D0–D1/negligible damage in the three inventories. In this study, the D5–D6 damage in the AIJ-data and the major damage in the V-data and the BRI-data were defined as collapse as shown in Figure 2, and the D0–D1 damage and the negligible damage were defined as non-collapse for the following damage identification.

**Table 2.** Numbers of buildings in damage inventories.

| (a) AIJ-data (the 2016 Kumamoto Earthquake) | | | | | | | |
|---|---|---|---|---|---|---|---|
| | **D0** | **D1** | **D2** | **D3** | **D4** | **D5** | **D6** | **Total** |
| Number | 492 | 562 | 245 | 272 | 237 | 224 | 68 | 2100 |
| Percentage (%) | 23.4 | 26.8 | 11.7 | 13.0 | 11.3 | 10.7 | 3.2 | - |

| (b) V-data (the 2016 Kumamoto earthquake) | | | | |
|---|---|---|---|---|
| | **Negligible** | **Moderate** | **Heavy** | **Major** | **Total** |
| Number | 6014 | 1402 | 439 | 318 | 8173 |
| Percentage (%) | 73.6 | 17.2 | 5.4 | 3.9 | - |

| (c) BRI-data (the 1995 Kobe earthquake) | | | | |
|---|---|---|---|---|
| | **Negligible** | **Moderate** | **Heavy** | **Major** | **Total** |
| Number | 21,005 | 12,721 | 5966 | 6782 | 46,474 |
| Percentage (%) | 45.2 | 27.4 | 12.8 | 14.6 | - |

## 2.3. Aerial Images and Extraction of Blue Tarp-Covered Buildings

Aerial images observed immediately after the earthquakes were analyzed in this study. We used the aerial images in Mashiki town observed on 19 April 2016, three days after the Kumamoto earthquake, and the aerial images in Nishinomiya city observed on 21 January 1995, four days after the Kobe

earthquake. The images consist of three visible bands (RGB), and they were ortho-rectified to 8-bit images with the spatial resolution of 0.2 m.

Figure 4 shows the typical buildings for each damage grade in the 2016 Kumamoto and 1995 Kobe earthquakes. We can clearly identify rubbles in and around the severely damaged buildings such as D5, D6, and major damage levels in the aerial images. Some of the affected buildings, such as D2, moderate and heavy damage, are covered with blue tarps on their roofs as shown in the ground photographs. In order to analyze such blue tarp-covered buildings, buildings whose roofs are approximately 50% covered with blue tarps are visually identified from the aerial images. Since the detection of all the blue tarp buildings from whole images is time-consuming and labor-intensive work, certain numbers of blue tarp buildings were randomly extracted from the images. Table 3 shows the numbers of the extracted blue tarp buildings for each damage grade of the damage inventories. In the AIJ-data, approximately 65 % of the blue tarp buildings were classified as D2–D4. In the V-data and BRI-data, 80–90% of the blue tarp buildings were classified as moderate or heavy damage. These results indicate that the blue tarps on building roofs can be judged as a typical sign of intermediate damage grade. Furthermore, it is more feasible to extract the blue tarp buildings than to classify all the intermediate damage grades. Therefore, we tried to classify the damage grades of the buildings to three categories; collapsed (without blue tarp), non-collapsed (without blue tarp) and blue tarp-covered buildings.

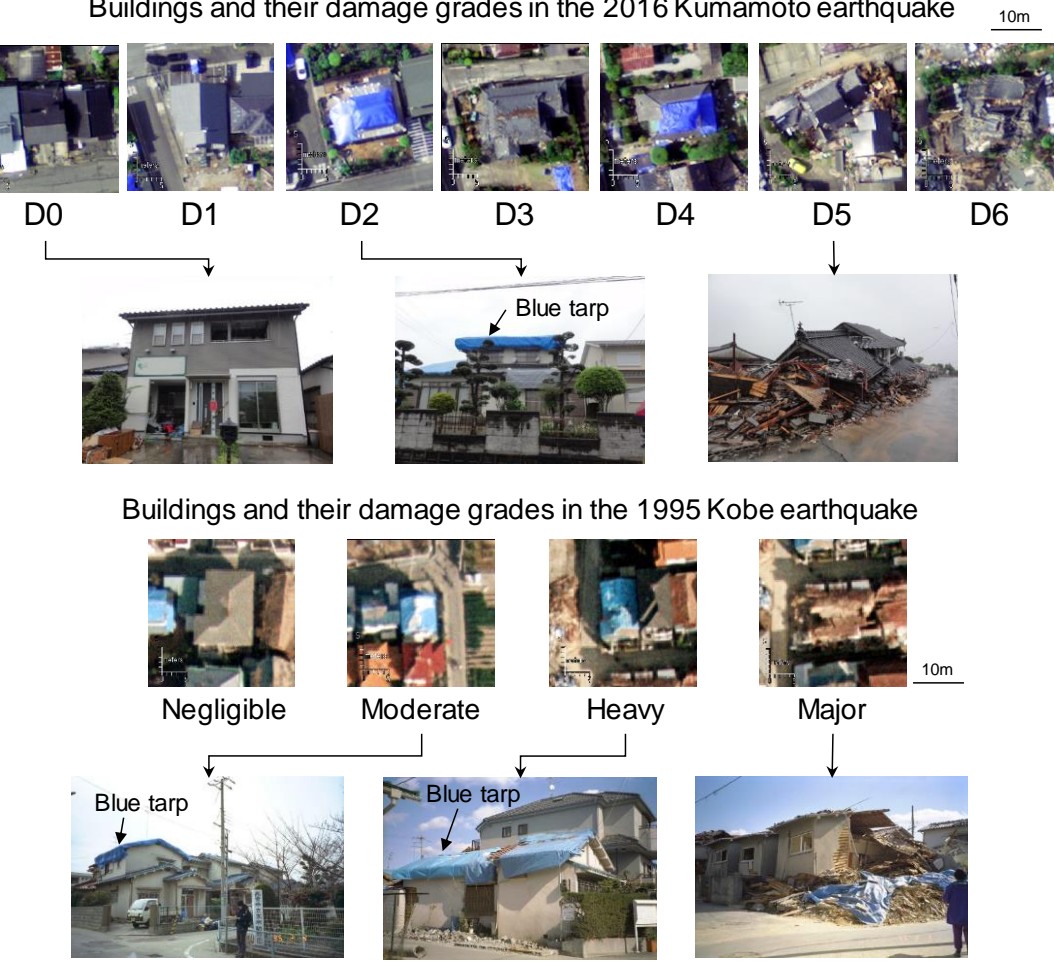

**Figure 4.** Close-ups of the typical buildings in the aerial images for each damage grade with ground photographs. The ground photographs of the 1995 Kobe earthquake are derived from the Nishinomiya database [36].

**Table 3.** Damage grades of extracted blue tarp covered buildings.

| **(a) AIJ-Data** | | | | | | | | |
|---|---|---|---|---|---|---|---|---|
| | **D0** | **D1** | **D2** | **D3** | **D4** | **D5** | **D6** | **Total** |
| Num. of blue tarp covered buildings | 1 | 64 | 76 | 47 | 16 | 5 | 2 | 211 |
| Percentage (%) | 0.5 | 30.3 | 36.0 | 22.3 | 7.6 | 2.4 | 0.9 | - |

| **(b) V-data** | | | | | |
|---|---|---|---|---|---|
| | **Negligible** | **Moderate** | **Heavy** | **Major** | **Total** |
| Num. of blue tarp covered buildings | 13 | 287 | 88 | 17 | 405 |
| Percentage (%) | 3.2 | 70.9 | 21.7 | 4.2 | - |

| **(c) BRI-data** | | | | | |
|---|---|---|---|---|---|
| | **Negligible** | **Moderate** | **Heavy** | **Major** | **Total** |
| Num. of blue tarp covered buildings | 176 | 755 | 387 | 147 | 1465 |
| Percentage (%) | 12.0 | 51.5 | 26.4 | 10.0 | - |

## 3. Methodology

### 3.1. Preprocessing

In order to develop a deep-learning-based damage-classification model, training and validation samples were prepared from the damage inventories. Since even the damage inventories included over- or under-interpretations in the damage assignment as discussed later, we carefully selected the building data for accurate classification. Table 4a shows the numbers of buildings selected for the training and validation data from each inventory. Approximately 1300, 2200 and 5700 buildings were extracted from the AIJ-data, V-data and BRI-data, respectively. It was difficult to determine an ideal ratio to divide the data into training and validation sets. Since the total number of samples was limited, a larger number of samples was given to the training dataset. In this study, 80% of the randomly selected samples was used as training data in the following hold-out validation, and the remaining 20% were classified as validation data as shown in Table 4a.

**Table 4.** (**a**) Numbers of the original training and validation data. (**b**) Numbers of the training and validation data after the data augmentation.

| **(a) Original** | | | | | | |
|---|---|---|---|---|---|---|
| | **AIJ-data** | | **V-data** | | **BRI-data** | | **Total** |
| | **Training** | **Validation** | **Training** | **Validation** | **Training** | **Validation** | |
| Non-collapsed | 683 | 171 | 1270 | 318 | 2892 | 723 | 6057 |
| Blue tarp covered | 169 | 42 | 324 | 81 | 1171 | 294 | 2081 |
| Collapsed | 158 | 39 | 156 | 39 | 466 | 116 | 974 |
| Total | 1010 | 252 | 1750 | 438 | 4529 | 1133 | 9112 |
| **(b) After data augmentation** | | | | | | |
| | **AIJ-data** | | **V-data** | | **BRI-data** | | **Total** |
| | **Training** | **Validation** | **Training** | **Validation** | **Training** | **Validation** | |
| Non-collapsed | 683 | 171 | 1270 | 318 | 2892 | 723 | 6057 |
| Blue tarp covered | 507 | 126 | 972 | 243 | 3366 | 843 | 6057 |
| Collapsed | 1003 | 250 | 991 | 250 | 2851 | 712 | 6057 |
| Total | 2193 | 547 | 3233 | 811 | 9109 | 2278 | 18,171 |

Before applying the CNN-based classification, image patches of the buildings were generated from the aerial images and the building inventories. Figure 5 shows the flow of the preprocessing adopted in this study. The building locations were identified from the building footprints in the inventories. Square-shaped image patches created from the images with the guide of the building footprints by adopting a larger length for the north–south or east–west directions of the buildings.

A buffer area with the length of 15 pixels (3 m) was given to the both sides of the patches as shown in Figure 5, because gaps are sometimes found between the building outlines in the images and the building footprints.

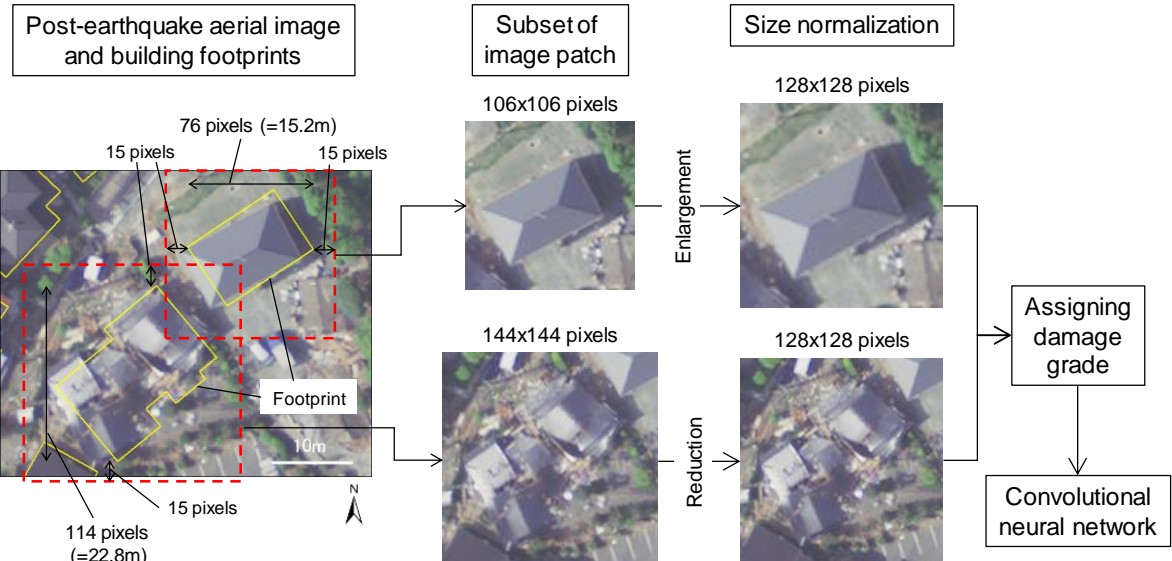

**Figure 5.** Flow for the preprocessing of the image patches extracted from the aerial image.

Since image patches with fixed numbers of pixels were analyzed in the following CNN-based approach, the size of the image patches needed to be normalized. To determine the size of the image patches, the building size in the inventories was analyzed. Figure 6 shows the relative and cumulative frequency distributions of the building size in a square meter derived from each damage inventory. The most frequent building size was around 100 square meters in the AIJ-data and V-data, and 50 square meters in the BRI-data. The sizes of approximately 95 % of the buildings were smaller than 300 square meters. The equivalent length and width of a building with the size of 300 square meters were about 17 m, corresponding to 87 pixels in the aerial images. Considering the orientation of the buildings on the images, the size of the image patches needs to be larger than 87 pixels. Excessively large image sizes should not be used to suppress the over-enlargement of the image patches. A power of two has been widely adopted for image dimensions in traditional image processing such as texture analysis. We determined the size of the image patches as $128 \times 128$ pixels, because 128 was a number of two to the seventh power and nearest larger number to 87 pixels. All the extracted image patches were normalized to $128 \times 128$ pixels by digitally enlarging and reducing the sizes for the following analysis.

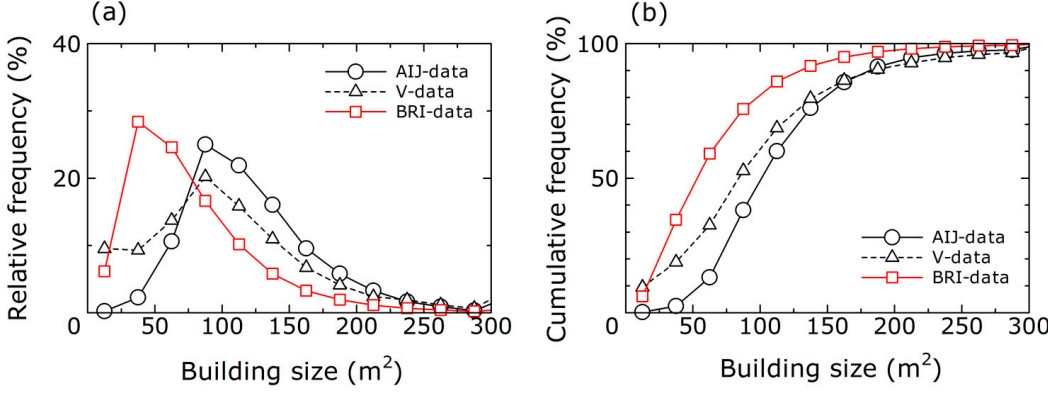

**Figure 6.** (**a**) Relative frequencies of the building size. (**b**) Cumulative frequencies of the building size.

### 3.2. Development of Convolutional Neural Network (CNN) Model

In this study, the CNN architecture and hyperparameters developed in the previous damage detection study by one of the authors [37] were tuned and improved by checking the applicability of the models to our datasets through trial and error. Figure 7 shows the CNN architecture adopted for damage classification in this study. Table 5 shows the adopted kernel size and size of shape in each layer. The basic convolution block consists of a convolution layer, a nonlinear layer and a pooling layer. The input to the convolutional layer was a three-dimensional array with two-dimensional feature maps of size $m \times m$ and the number of spectral bands $r$ (here, $m \times m \times r = 128 \times 128 \times 3$). The convolutional layer has $k$ trainable filters of kernel size $l \times l$, which connects the input feature map to the output feature map. In the nonlinearity layer, a rectified linear unit (ReLU) was chosen to compute the feature maps. The pooling layer served to progressively reduce the spatial size of the features, to reduce the number of parameters and the amount of computation in the network, and hence to also control the overfitting. The Max pooling was adopted in the layers. Four basic convolution blocks were included in the CNN model, and consequently the original image patch was transformed into a feature map with a size of $3 \times 3 \times 128$.

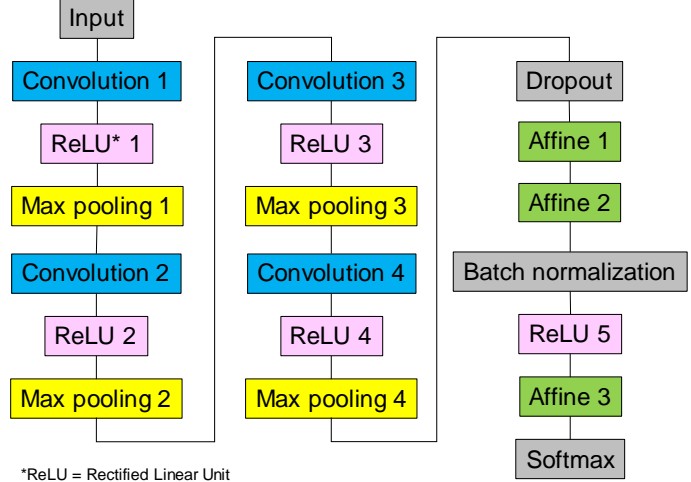

**Figure 7.** Convolutional neural network (CNN) architecture adopted in this study.

**Table 5.** Kernel and shape size adopted in the CNN model.

| Layer | Kernel Size | Shape (Length × Width × Bands) | Layer | Kernel Size | Shape (Length × Width × Bands) |
|---|---|---|---|---|---|
| Input | | $128 \times 128 \times 3$ | Convolution 4 | 2 | $6 \times 6 \times 128$ |
| Convolution 1 | 6 | $62 \times 62 \times 16$ | ReLU 4 | | $3 \times 3 \times 128$ |
| ReLU 1 | | $31 \times 31 \times 16$ | Max Pooling 4 | 2 | $3 \times 3 \times 128$ |
| Max Pooling 1 | 2 | $31 \times 31 \times 16$ | Dropout | | |
| Convolution 2 | 2 | $30 \times 30 \times 32$ | Affine 1 | | 1152 |
| ReLU 2 | | $15 \times 15 \times 32$ | Affine 2 | | 576 |
| Max Pooling 2 | 2 | $15 \times 15 \times 32$ | Batch normalization | | |
| Convolution 3 | 2 | $14 \times 14 \times 64$ | ReLU 5 | | 576 |
| ReLU 3 | | $7 \times 7 \times 64$ | Affine 3 | | 3 |
| Max Pooling 3 | 2 | $7 \times 7 \times 64$ | Softmax | | 3 |

Dropout and batch normalization (BN) were also employed in the fully connected layers. Dropout was introduced in the fully connected layer in order to suppress overfitting by removing the randomly selected neurons with the probability of 0.5 at the training stage [38]. The BN was recognized as a powerful tool for suppressing overfitting and accelerating convergence speed [39]. In the BN, the input values within a mini-batch are normalized to the mean of zero and the variance of 1 to correct the distribution of each layer input. In the affine layers, the contained feature maps were transformed

into output nodes by the linear combination of the nodes of the previous layer with an added bias. After the affine transformations and BN in the fully connected layers, the final output dimension was converted into three classes for the softmax function as shown in Table 5. The size of the mini-batch was set to 100 to control the number of training samples in an epoch. Since we confirmed that the loss functions for the training and validation samples were not significantly reduced under the number of epochs larger than 1000 in our preliminary analysis, the number of epochs of 1000 was adopted in the training process.

Since the number of training data was limited as shown in Table 4a, data augmentation was performed by horizontally and vertically flipping, rotating and adding contrast to the original image patches. The number of training and validation data is shown in Table 4b. The convergence of the loss function for the training and validation data by the developed CNN model is illustrated in Figure 8. We confirmed that overfitting was not observed during the learning process and the learning is successfully performed by the developed model. The classification accuracy for the validation data is represented in a confusion matrix shown in Table 6. The numbers in the shaded cells indicate the number of buildings correctly classified by the model. The precisions and recalls for all the damage grades are higher than 97%, indicating that the damage grades were successfully classified by the model.

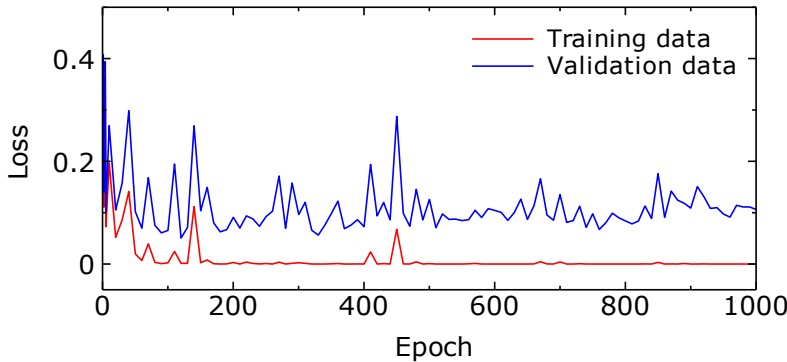

**Figure 8.** Convergence of the loss functions for the training and validation data.

**Table 6.** Classification accuracy for the validation data.

| | | Damage Inventory | | | | Recall (%) |
| | | Non-Collapsed | Blue Tarp Covered | Collapsed | Total | |
|---|---|---|---|---|---|---|
| | **Non-collapsed** | 1184 | 14 | 3 | 1201 | 98.6 |
| **Estimated** | Blue tarp covered | 7 | 1190 | 4 | 1201 | 99.1 |
| | Collapsed | 21 | 8 | 1205 | 1234 | 97.6 |
| | Total | 1212 | 1212 | 1212 | 3636 | |
| Precision (%) | | 97.7 | 98.2 | 99.4 | | |
| Overall accuracy (%) = | | | | | | 98.4 |

## 4. CNN-Based Building Damage Estimations

The developed CNN model was applied to the whole building data. Figure 9a,b show the distributions of the estimated building damage for the Mashiki town and Nishinomiya city, respectively. Compared with the damage inventories shown in Figure 3a,b, the D5–D6/major damage buildings were mostly detected as the collapsed buildings by the CNN model. However, the distribution of the collapsed buildings seems to be overestimated since the D0–D1/negligible damage buildings were falsely classified to the collapsed buildings. The blue tarp-covered buildings were distributed mainly between the collapsed and non-collapsed building areas. A similar trend was also confirmed in the damage inventory because the D2–D4, moderate and heavy damage buildings, were located around the D5–D6/major damage buildings in the damage inventories.

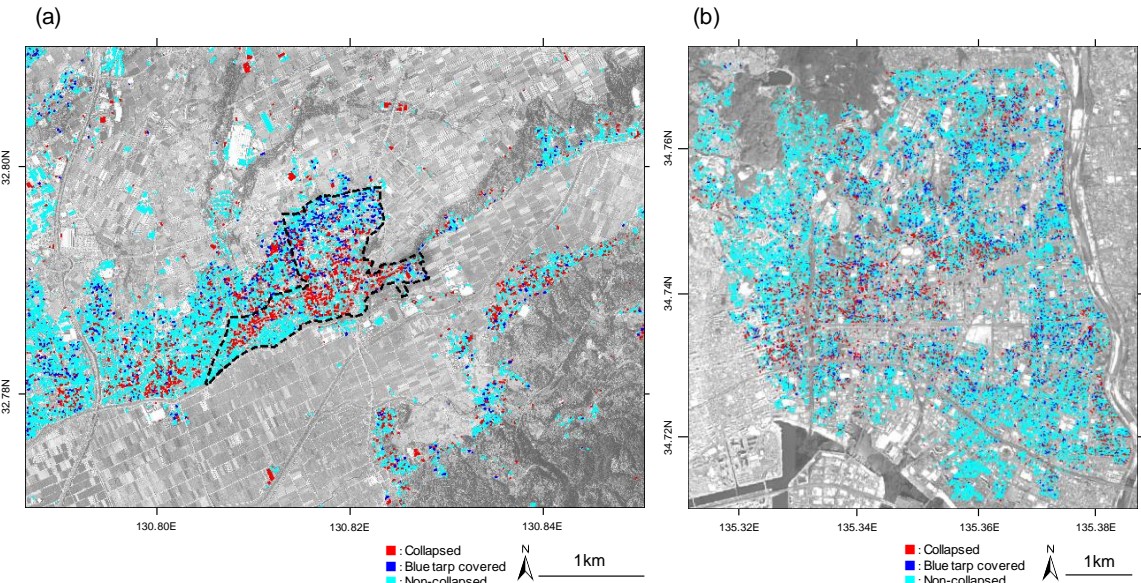

**Figure 9.** Damage distributions estimated by the CNN model for (**a**) Mashiki town in the 2016 Kumamoto earthquake and (**b**) Nisinomiya city in the 1995 Kobe earthquake.

Table 7 shows the summaries of the CNN-based classifications. The numbers of the cells indicate the number of buildings classified to each class by the CNN model. For the accuracy assessment, the colors of the cells were classified into three categories; underestimated (light blue), correctly estimated (light green) and overestimated (pink) as shown in the table. The precisions and recalls were calculated for each damage class. The precisions for the non-collapsed and collapsed buildings were higher than 75% in the AIJ- and V-data. Whereas the recalls for the non-collapsed buildings were higher than 70% in the AIJ- and V-data, the recalls for the collapsed buildings were lower than 40% in the three inventories, indicating that the overestimations were found in the collapsed building classification. Although the blue tarp-covered buildings do not perfectly correspond to moderate or heavy damage, the precision and recall were indicated with brackets. The recalls for the blue tarp buildings were higher than 65%, but the precisions were lower than 30 %. This is because not all of the intermediately damaged buildings were covered with blue tarps on the roof, whereas the blue tarp buildings were typically assigned to the intermediate damage level.

The spatial distribution of the difference between the estimations and the inventories is visualized in Figure 10a,b. The colors of the polygons indicate the results of the accuracy assessment classified in Table 7. For the 2016 Kumamoto earthquake shown in Figure 10a, the correctly estimated buildings were dominantly distributed in the area. It indicates that the building damage was accurately estimated by the proposed model. For the 1995 Kobe earthquake, shown in Figure 10b, on the other hand, not only the correctly estimated buildings but also the underestimated buildings were widely distributed. This is mainly because the numbers of the moderately damaged buildings were classified as non-collapsed as shown in Table 7. As indicated in Figure 2, moderate damage in the BRI-data included slight damage levels, such as hair-line cracks in walls. Since it is very difficult to detect such slight damage from the aerial images, these buildings were underestimated in the proposed model. Considering the limitation of the remote-sensing-based damage detection, these underestimations would be acceptable for a quick damage assessment.

The causes of other serious false classifications are discussed by carefully checking the images and the building damage data. Figure 11a shows examples of D5 buildings in the AIJ-data falsely classified to non-collapse by the CNN model. The buildings were assigned to collapse in the inventory because soft-story collapse was confirmed in the field investigation. In the aerial images, however, failures were not found on the building roofs. When we carefully see the images, we can find rubbles of columns and walls distributing around the buildings. It is difficult to detect such soft-story collapse of buildings

without failure of roofs from vertical images even by visual interpretation since the building roofs seem to be intact in the images. Therefore, the damage grades were underestimated by the CNN model. On the other hand, Figure 11b,c shows examples of D0–D1 buildings falsely classified as collapsed by the CNN model. We can see that the roofs of the buildings in Figure 11b are seriously damaged. Considering the criteria of the damage grades in Table 1 and Figure 2, the damage level for the buildings should be classified to D3 or higher. This suggests that the damage was probably falsely labeled in the field investigation due to missing the damaged parts. Due to the increase in solar energy generation in residential houses in Japan, solar panels have been installed on the roofs of many buildings. Figure 11c shows the examples of the building roofs with solar panels that were not seriously damaged by the earthquake and assigned to D0 or D1 in the damage inventory. The damage of the buildings, however, was classified to collapse by the CNN model. As shown in Figure 11c, the color and contrast of the solar panels were totally different from those of the roof materials. The damage of such buildings was overestimated as collapsed, probably because the solar panels were falsely interpreted as damage in the CNN model. These results show the limitation of the proposed damage-identification method especially for buildings subjected to soft-story collapse and buildings with solar panels of the roofs. Although it would be difficult to accurately detect soft-story collapse from the remote-sensing images, the overestimation caused by the presence of solar panels would be reduced by increasing training samples in our future researches.

**Table 7.** Result of the CNN-based damage classification for each damage inventory. The colors of the cells indicate the labels of the classification; correctly estimated (light green), underestimated (light blue) and overestimated (pink). Accuracies for the blue tarp-covered buildings are shown by brackets since the blue tarp does not perfectly correspond to D2–D4/moderate, heavy damage levels.

**(a) AIJ-data**

|  |  | Damage inventory | | | | | | | Total | Recall (%) |
|---|---|---|---|---|---|---|---|---|---|---|
|  |  | D0 | D1 | D2 | D3 | D4 | D5 | D6 |  |  |
| Estimated | Non-collapsed | 465 | 444 | 98 | 115 | 102 | 32 | 3 | 1259 | 72.2 |
|  | Blue tarp-covered | 4 | 74 | 89 | 55 | 20 | 6 | 2 | 250 | (65.6) |
|  | Collapsed | 23 | 44 | 58 | 102 | 115 | 186 | 63 | 591 | 42.1 |
|  | Total | 492 | 562 | 245 | 272 | 237 | 224 | 68 | 2100 | - |
| Precision (%) |  | 86.2 | | (21.8) | | | 85.3 | | - | - |

**(b) V-data**

|  |  | Damage inventory | | | | Total | Recall (%) |
|---|---|---|---|---|---|---|---|
|  |  | Negligible | Moderate | Heavy | Major |  |  |
| Estimated | Non-collapsed | 5049 | 752 | 126 | 47 | 5974 | 84.5 |
|  | Blue tarp-covered | 118 | 387 | 124 | 20 | 649 | (78.7) |
|  | Collapsed | 847 | 263 | 189 | 251 | 1550 | 16.2 |
|  | Total | 6014 | 1402 | 439 | 318 | 8173 | - |
| Precision (%) |  | 84.0 | (27.8) | | 78.9 | - | - |

**(c) BRI-data**

|  |  | Damage inventory | | | | Total | Recall (%) |
|---|---|---|---|---|---|---|---|
|  |  | Negligible | Moderate | Heavy | Major |  |  |
| Estimated | Non-collapsed | 18,154 | 9335 | 3798 | 3527 | 34,814 | 52.1 |
|  | Blue tarp-covered | 634 | 1914 | 1057 | 556 | 4161 | (71.4) |
|  | Collapsed | 2217 | 1472 | 1111 | 2699 | 7499 | 36.0 |
|  | Total | 21,005 | 12,721 | 5966 | 6782 | 46,474 | - |
| Precision (%) |  | 86.4 | (15.9) | | 39.8 | - | - |

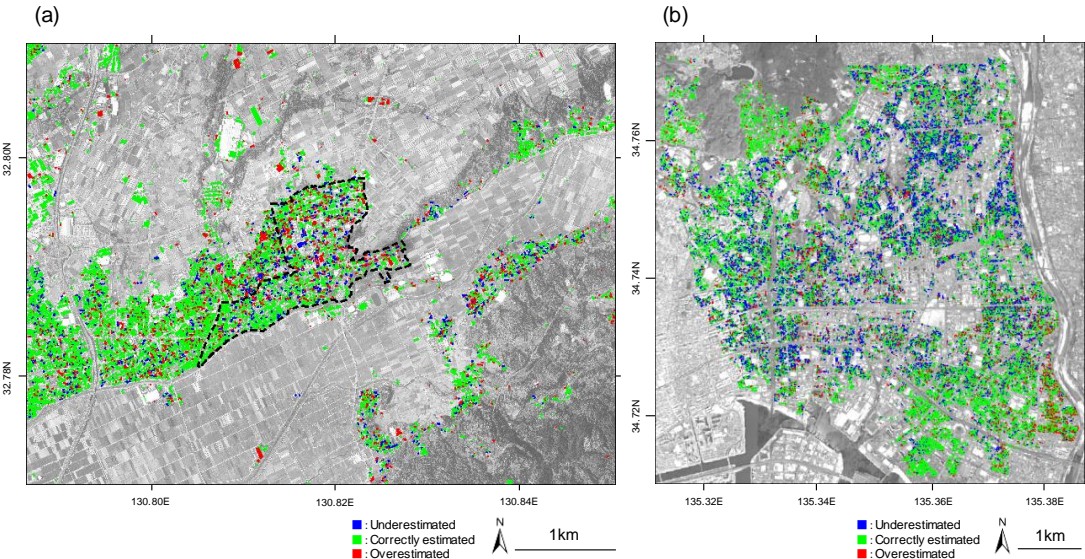

**Figure 10. (a)** 2016 Kumamoto earthquake; **(b)** 1995 Kobe earthquake. Distributions of the difference of damage classes between the estimation and the inventory. The green regions indicate the buildings correctly classified by the CNN model. The blue and red regions represent the under- and overestimated areas by the CNN model, respectively.

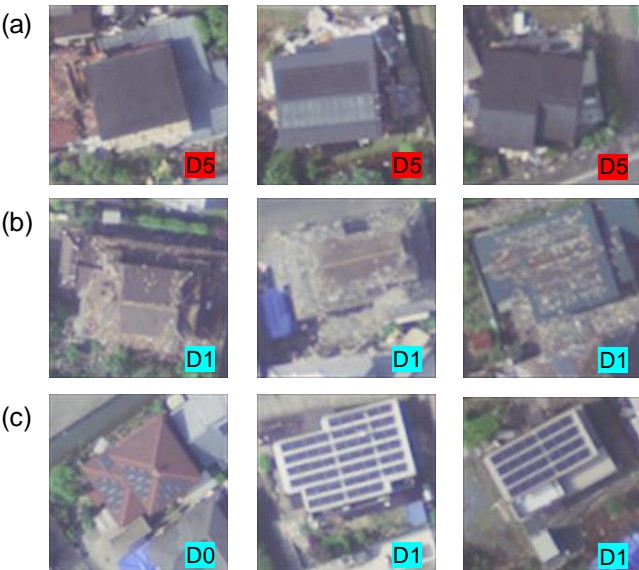

**Figure 11.** Close-ups of the images falsely classified by the CNN model. **(a)** Soft-story collapsed buildings (D5). **(b)** D1 buildings with seriously damaged roofs. **(c)** D0–D1 buildings with solar panels on the roofs.

## 5. Application to Buildings Damaged by a Typhoon

In 9 September 2019, the typhoon Faxai attacked the eastern part of Japan. Coastal areas in Chiba prefecture (see Figure 1a) were severely damaged by the strong winds of the typhoon, and the number of the affected buildings exceeded 65,000 in the Chiba prefecture. Kyonan town, located in the western coastal area of the Chiba prefecture, was one of the severely damaged areas by the typhoon. Figure 12 shows an aerial photograph captured at the town from a helicopter immediately after the typhoon [40]. Since the building roofs were seriously damaged by the strong winds, we can find numbers of building roofs being covered with blue tarps.

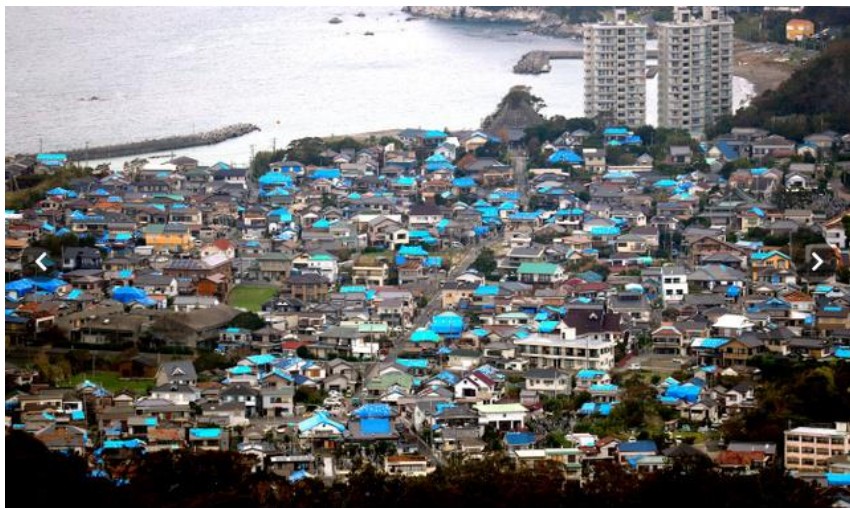

**Figure 12.** Aerial photograph of Kyonan town captured on 14 September 2019 [40].

The damage of the buildings was generated by the vibrations of ground, foundations and superstructures in earthquake disasters. Buildings are damaged by the unilateral force of winds in typhoon disasters. Whereas most elements of buildings including foundations are affected by earthquakes, the outer parts of buildings, including roofs and walls, were mainly damaged by typhoons. Although the failure mechanism of wind-induced building damage was slightly different from that of an earthquake-induced damage, the damage classification of the BRI-data shown in Table 1 has been widely used in Japan, also for building damage assessments in typhoon disasters. This means that the proposed damage-identification method would be applicable, not only for earthquake disasters, but also for typhoon disasters.

We apply the proposed CNN model to aerial images in Kyonan town. Figure 13a shows the aerial image observed eighteen days after the typhoon. The target area covers the length of 3 km for the north–south direction and the width of 4 km for the east–west direction, respectively. Since the building damage inventory in this town was not available, the locations and damage grades of the buildings were visually interpreted by the authors. The damage grades were classified into three categories; collapsed, blue tarp-covered and non-collapsed. Figure 13a also shows the distribution of the interpreted building damage. The target area included approximately 2300 buildings in total, and 18 and 729 buildings were classified as collapsed and blue tarp-covered, respectively.

The image patches were extracted from the aerial images, and the building damage for each image patch was estimated by the proposed CNN model. Figure 13b shows the distribution of the building damage estimated by the model. The distribution of the estimated blue tarp buildings show good agreement with that of the visually interpreted ones. Table 8 summarizes the classification accuracies of the CNN model. The result shows that more than 90% of the buildings are correctly classified. The severely damaged buildings, however, were slightly overestimated by the CNN model. The damage level was overestimated when rubbles or small objects were distributed around the buildings, and solar panels were installed on the building roof as shown in Figure 11c. Whereas such overestimations were found in some buildings, we confirmed that the proposed CNN worked well for the identification of building damage, not only by the earthquakes but also by the typhoon. The methodology introduced in this study would be useful for the rapid identification of damage distribution, only from post-disaster aerial images.

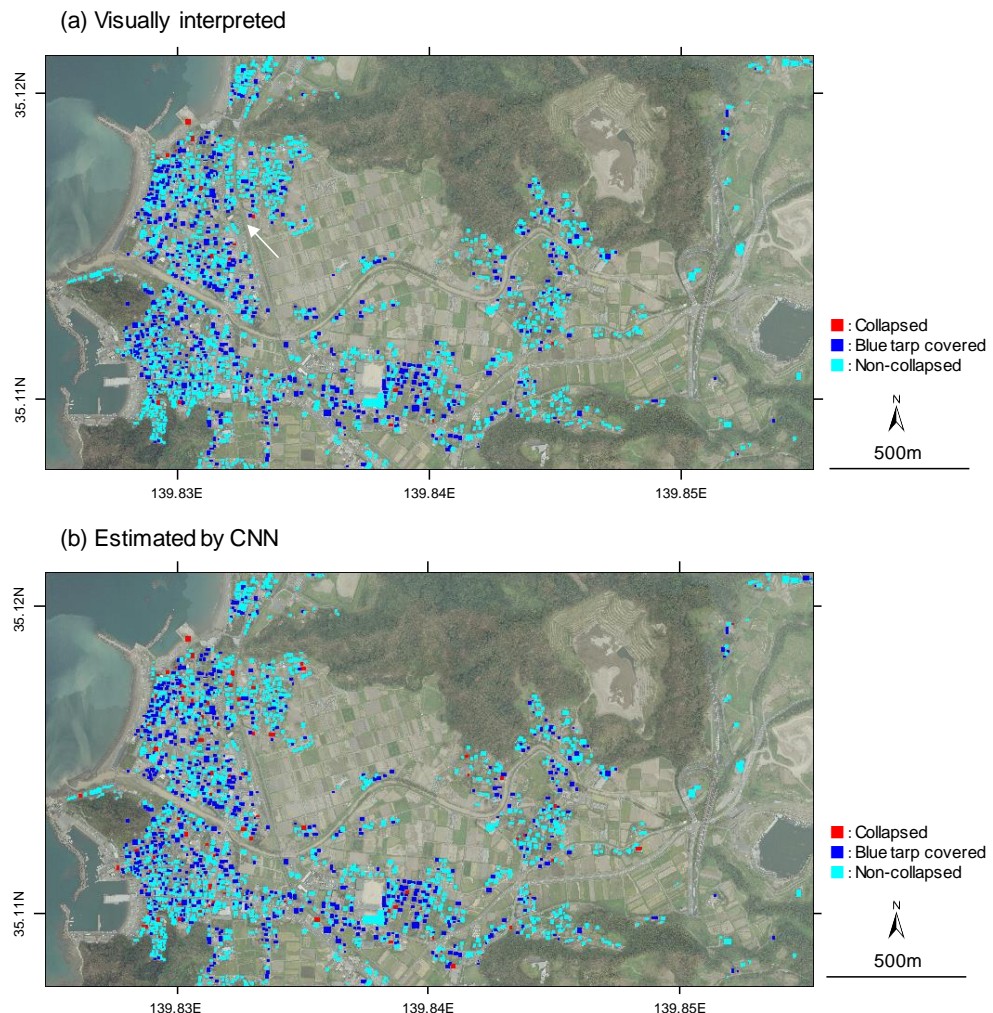

**Figure 13.** (**a**) Visually interpreted building damage in Kyonan town. The arrow indicates the approximate direction of the aerial photograph in Figure 12. (**b**) Building damage estimated by the proposed CNN model.

**Table 8.** Classification accuracies of the building damage in Kyonan town.

| | | Visually Interpreted | | | | Recall (%) |
|---|---|---|---|---|---|---|
| | | Non-Collapsed | Blue tarp-Covered | Collapsed | Total | |
| **Estimated** | Non-collapsed | 1469 | 17 | 2 | 1488 | 98.7 |
| | Blue tarp-covered | 57 | 703 | 1 | 761 | 92.4 |
| | Collapsed | 61 | 9 | 15 | 85 | 17.6 |
| | Total | 1587 | 729 | 18 | 2334 | |
| Precision (%) | | 92.6 | 96.4 | 83.3 | | |
| Overall accuracy (%) = | | | | | | 93.7 |

## 6. Conclusions

In this study, the convolutional neural network (CNN)-based methodology for identifying the building damage from post-disaster aerial images was developed using the building damage inventories and the aerial images obtained in the 2016 Kumamoto, and the 1995 Kobe, Japan earthquakes. The building damage was classified into three categories; collapsed, non-collapsed and blue tarp-covered buildings. Considering the damage classification in the inventories, collapse and non-collapse were defined as D5–D6/major damage and as D0–D1/negligible damage, respectively. We confirmed that blue tarp-covered buildings predominantly represented an intermediate damage level such as D2–D3/moderate damage.

The CNN model was developed based on the training and validation samples extracted from the aerial images and the damage inventories. The CNN architecture consisted of four feature extraction steps by the convolutional, nonlinearity and pooling layers. Dropout and batch normalization were also employed in the CNN model. The result for the validation data showed that more than 97% of the buildings were correctly classified by the CNN model. The model was applied to all the building data to estimate the damage distributions in the 2016 Kumamoto and 1995 Kobe earthquakes. Whereas some discrepancies of damage grades were found between the inventories and the estimations, the estimated damage distributions showed agreement with those in the damage inventories.

The proposed CNN model was also applied to the aerial images in Kyonan town, Chiba prefecture, affected by the typhoon in September 2019. The locations and damage grades were visually interpreted. The estimated damage distribution shows good agreement with the visually interpreted one. The accuracy assessment revealed that approximately 94% of the buildings were correctly classified by the CNN model. These results indicated that the proposed CNN model would be useful for the rapid identification of damage distribution using post-disaster aerial images.

We found the over- and underestimations of our results because it was difficult to discriminate from small features around the buildings and solar panels on building roofs from building rubbles, and to identify soft-story collapse from the images. It was also difficult to accurately classify negligible and moderate damage such as hair-line cracks in walls. Although we confirmed the limitations of remote-sensing-based damage detections, we intended to develop more a accurate classification technique by increasing the appropriate training samples such as soft-story collapse and solar panels on the roofs.

Finally, we proposed a considerable framework for the quick and efficient damage assessment including remote-sensing-based damage detections. In actual post-disaster responses, firstly obtained damage information needs to be continuously updated when new damage information was gathered from other data sources. Our proposed method would be useful in the first phase of the damage assessment because it can provide damage distribution building by building, more rapidly than visual interpretation. Estimated damage maps could be updated by visually screening the estimated building damage and by additionally classifying the moderate/heavy damage levels from visual interpretations in the second phase. The damage maps could be further updated by including detailed damage information obtained in field investigations at the third phase. Such multi-phased damage assessment could provide more reliable damage maps for efficient disaster responses.

**Author Contributions:** H.M. conceived and designed the experiments and wrote the manuscript; and T.A. performed the experiments and analyzed the data. M.M. supervised the experiments and provided the correction of the manuscript. All authors have read and agreed to the published version of the manuscript.

**Funding:** This research was funded by the Japan Society for the Promotion of Science (KAKENHI Grant numbers 19H02408).

**Acknowledgments:** The authors thank the Kyushu Branch of Architectural Institute of Japan for providing the building damage data, and PASCO Corporation for providing the aerial images and the visual interpretation data of the 2016 Kumamoto earthquake. The authors are also grateful to Aero Asahi Corporation for providing the aerial images in Kyonan town, Chiba prefecture.

**Conflicts of Interest:** The authors declare no conflict of interest.

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
