# Peer review of "Deep Learning-Based Identification of Collapsed, Non-Collapsed and Blue Tarp-Covered Buildings from Post-Disaster Aerial Images"

_remotesensing, doi:10.3390/rs12121924_

Round 1
Reviewer 1 Report
I have completed my review of the manuscript, ‘Deep Learning-based Identification of Collapsed, Non-collapsed and Blue Tarp Covered Buildings from Post-Disaster Aerial Images’ by Hiroyuki Miura et al. In this contribution, the authors present a new methodology for automated identification of building damage from post-disaster aerial, which disaster including earthquake and typhoon. However, in its present form, the analysis of this dataset seems very cursory, and while the conclusions drawn from it are interesting, it’s hard to evaluate whether they are warranted based on the level of analysis presented. I highlight my concerns in more detail below. In conclusion, I encourage the authors to work on the robustness of their analysis. This is currently needed to allow this study to be published.
Basically, I think that the case study findings are interesting. However, the introduction is cast too broadly in terms of the development of CNN without a compelling and obvious link. It should take more space on the technical aspects of CNN rather than extensively discuss traditional image interpretation. Plus, this is a change detection study and I was really hoping to see a map of differencing between estimation and observation to spatially visualize the patterns described in the study. Moreover, poor or incomplete description of CNN methods.
Line-by-line comments:
L86 I ’m not quite sure that this study did some work in this section and that those work are references. Maybe divide into more segments, such as study sites and background information, and research materials could help.
L89-92 I don’t quite understand the seismic intensity distribution described in this paragraph. That meaningful information should be added to FIg.1.
Figure 1. The location map isn’t very helpful because it is difficult to see the location of the study site. Plus, the blank space in the upper left corner looks awkward.
Figure 2. Although I can see the slightly difference between D0 and D1 on the wall, and however, this is not very obvious, maybe different colors or thicker lines will help. I didn’t see any information about AIJ-data (a very little in L93), BRI-data and V-data in the previous paragraphs, and they suddenly appeared in Fig. 2.
L104 please define “Normal” buildings.
Figure 3. Excessively large legends, north arrows and scales obscure meaningful information. Overall, this figure looks very unprofessional.The aerial images, for example, figure 4 needs scale bar.
L166 the ratio of training data and verification data will affect the performance of the model, so I hope to see how the author decides this ratio.
L175-181 I read this paragraph five times, but I still don't know how to calculate the values in figure 6.
Figure 6(b). V-data disappeared.
L223 I hope to see more discussions about the poor performance of Mode in Moderate or Heavy damage (from 15.9% to 27.8%).
Figure 9 did not provide the meaningful information.
L250 I am not a civil engineering expert, but I imagine that the collapse of houses caused by earthquakes and typhoons may be different. Will this difference cause interpretation errors? I hope to see more discussions about this. This will also help us to consider the factor of applying this technology later.
Author Response
Thank you for your critical reviews and comments. Our responses to your comments are shown in the uploaded file.

Reviewer 2 Report
The paper is well written and clear, despite the presence of some minor typos and grammar errors, especially concerning verbs and sentence construction. I recommend a careful revision of the english (line 27 page 1 - "have produced" and not "have been produced", line 40 page 1 ("approaches have" and not "approaches has"... to cite some.
It is not clear what the Authors mean by "Normal" buildings (they are introduced at page 3 line 104).
I understand that the study refers to a homogeneous class of buildings, essentially in terms of roof, thus excluding RC and non-residential buildings, but a sligthly more extensive description of the class of structure analysed could be useful not only for the readers that are not familiar with this typology but also to better understand the collapse mechanisms that the classification adopted (either EMS, AIJ, V, BRI or collapsed/non collapsed/blue tarp covered) somehow captures.
The limitations connected to soft floor mechanisms or the alterations caused by the presence of solar panels are clearly stated by the authors.
I encourage the authors to make an effort, maybe in the conclusions, to summarize the range of applicability of this methodology and to define an application protocol in which, for example, this method is combined (and needs to be so) with, for example visual inspections or other survey. This could frame the work done in real applications.
Author Response

(The authors gave the same response as above.)

Reviewer 3 Report
The authors present a very interesting paper where a Convolutional Neural Network is used to automatically identify the damage of buildings after a disaster. This is very relevant for civil protection purpouses.
The paper is well-structured, they use good data, and the method is well presented. I thank the authors for presenting Figure 2, where a comparison between the damage grades of Japan and Europe is done.
However, I have two major concerns that the author must solve before publication:
Query 1. A discussion on the hyper-parameters selection is missing. In this sense, the authors must justify the election. Have they performed a grid or random search or any other metaheuristic to select the values? This is a very important point that needs clarification.
Query 2. The use of English must be improved and should be reviewed by a native English professionar reviewer. For example:
- Line 27. "Large-scale natural disasters such as earthquakes have been produced a huge number of..." should be "Large-scale natural disasters such as earthquakes have produced a huge number of...".
- Line 31. "source to provide timely data for detection of damaged buildings for larger areas" should be "source to provide timely data for the detection of buildings damaged in large areas".
- And so on...
Author Response

(The authors gave the same response as above.)

Round 2
Reviewer 1 Report
I have completed my review of the revision, ‘Deep Learning-based Identification of Collapsed, Non-collapsed and Blue Tarp Covered Buildings from Post-Disaster Aerial Images’ by Hiroyuki Miura et al. In this version, the authors answer all (almost) of my concern in first review (eg. description of CNN methods, Comparable visual map, etc.). Therefore, I think that the study is quite ready for publishing in “remote sensing”.
Please correct the minor mistake.
Line-by-line comments:
Fig.1(c) A strange black line appears on the left of the map.
Author Response
We appreciate your comment. We modified Fig. 1(c) as shown in the modified manuscript.